# Experiences during Switching from Two-Stage to One-Stage Revision Arthroplasty for Chronic Total Knee Arthroplasty Infection

**DOI:** 10.3390/antibiotics10121436

**Published:** 2021-11-24

**Authors:** Guillem Navarro, Luis Lozano, Sergi Sastre, Rosa Bori, Jordi Bosch, Guillem Bori

**Affiliations:** 1Department of Traumatology and Orthopedic Surgery, Hospital Universitari Mútua Terrassa, 08221 Barcelona, Spain; guillemnaes@gmail.com; 2Department of Traumatology and Orthopaedic Surgery, Hospital Clínic de Barcelona, University of Barcelona, 08036 Barcelona, Spain; llozano@clinic.cat (L.L.); ssastre@clinic.cat (S.S.); rbori@clinic.cat (R.B.); 3Insitut d’ Investigacions Biomèdiques August Pi I Sunyer (IDIBAPS), 08036 Barcelona, Spain; 4Department of Microbiology, Hospital Clínic de Barcelona, University of Barcelona, 08036 Barcelona, Spain; jobosch@clinic.cat; 5Associate Professor of Traumatology and Orthopaedic Surgery, University of Vic-Central University of Catalonia (UVic-UCC), 08500 Vic, Spain

**Keywords:** 1-stage arthroplasty, single-stage revision, knee arthroplasty, knee infection

## Abstract

The objective of this study was to evaluate our preliminary results after changing our surgical strategy from 2-stage revision arthroplasty to 1-stage revision arthroplasty for patients with chronic knee periprosthetic joint infection. We conducted a prospective study of knee arthroplasty patients that had been diagnosed with chronic infection and treated using a 1-stage revision regardless of the traditional criteria applied for indication thereof. We evaluated two main variables: infection control and economic costs. The definitive diagnosis of infection of the revision was determined by using the criteria proposed by the Musculoskeletal Infection Society. The costs were calculated as average costs in USD, as described by Srivastava (2019), for 1-stage or 2-stage revisions. Eighteen patients were included in the study, and infection was controlled in 17 patients. The total economic savings for our hospital from these 18 patients amounted to USD 291,152. This clinical success has led to major changes in how our hospital approaches the treatment of chronically infected knee replacements, in addition to substantial economic advantages for the hospital.

## 1. Introduction

The treatment of chronic periprosthetic joint infection (PJI) is a source of considerable unease and confusion for arthroplasty surgeons. PJI remains a serious complication after total knee arthroplasty (TKA) and is one of the leading causes of TKA revision surgery. Currently, two approaches are generally used in the setting of chronic PJI of the TKA: 1-stage and 2-stage revision [1,2]. A 1-stage revision results in lower morbidity rates at the expense of a higher reinfection rate, while a 2-stage revision is exactly the opposite. Traditionally, 2-stage revision has been advocated for the treatment of chronic knee PJI [2,3]. This strategy requires, in the first stage, removal of infected components, an interim period with antibiotic treatment, and limited mobilization followed by a second stage with reimplantation of revision components. The reported success rates after 2-stage revision range from 80 to 95% [4]. However, high morbidity, mortality, and high economic costs often ensue [5,6,7]. In appropriate patients, a 1-stage revision is an alternative to a 2-stage revision and has a high success rate [8,9,10]. The 1-stage revision arthroplasty is an infrequent, underutilized surgical strategy, especially for chronic knee PJI [11]. In a meta-analysis by Kunutsor et al. [12], performed on 118 studies, 10 were 1-stage TKA studies and 108 were 2-stage TKA studies. In a recent comprehensive literature search of the last 20 years, Thakrar et al. [13] found 22 studies on 1-stage revision arthroplasty for PJI; 15 of them included PJI after total hip arthroplasty (THA), and only six included PJI after TKA, while a single study included both. Recently, several systematic reviews [10,12,14,15,16] concluded that there is no suggestion in the published studies that the two methods have different reinfection outcomes. However, in terms of functional outcomes [15] and quality of life [17], the models in these reviews do favor the 1-stage approach over the 2-stage approach.

The objective of this study was to evaluate preliminary results after changing our surgical strategy from 2-stage revision arthroplasty to 1-stage revision arthroplasty for patients with chronic knee PJI. The results evaluated were (1) infection control, and (2) economic costs after the reimplantation of the definitive arthroplasty.

## 2. Material and Methods

Up until September 2015, in Hospital Clínic de Barcelona, chronic knee PJI had been treated with 2-stage revision surgery. Studies in the past few years have indicated that (1) clinical results in terms of reinfection or chronic infection in 2-stage revision do not show great differences with the 1-stage revision [10,12,14,16]; (2) the results in terms of functional outcomes and quality of life have been shown to be better for 1-stage revision than for 2-stage revision [15,17]; (3) the results in our hospital for 1-stage THA revisions were better than for 2-stage THA revisions [18,19,20]; and (4) there was an organizational change in the orthopedic surgery and traumatology team, and chronic knee and hip PJI went from being treated in different departments to being treated by the same team of surgeons with experience in 1-stage and complex knee revision. This organizational change led our hospital to modify its treatment protocol for chronic knee PJI from the 2-stage approach to the 1-stage approach. After September 2015, all TKA in Hospital Clínic that were diagnosed with chronic PJI and not excluded for various reasons (Table 1) were treated by 1-stage revision. Treatment decisions from this date bypassed the traditional criteria used to determine the use of 1-stage revision [2,21,22], limiting the exclusion criteria as much as possible. We soon realized that there were cases where it was not possible to perform 1-stage knee revision arthroplasty, in contrast with chronic hip PJI, in which revision arthroplasty could be performed in all cases [19] (Table 1).

There were no exclusions on the basis of patient condition, American Society of Anesthesiologists classification, body mass index, soft tissue status (including the presence of a draining sinus), significant bone loss, failure of previous 1-stage or 2-stage procedures, unclear pre-operative bacterial specification (culture-negative preoperative aspiration) or causative microorganism including resistant organisms, or other patient-related factors potentially influencing the outcome. Cases in which 1-stage revision arthroplasty was not elected included patients coming from an acute infection in which recently performed debridement, antibiotics, and implant retention (DAIR) had failed, patients with systemic sepsis or large abscesses in the calf or along the entire limb, patients with significant skin defects who needed plastic surgery, patients with severe damage or rupture of the extensor mechanism, and fungal infection. In all these cases, a 2-stage revision was performed.

### 2.1. Patients

Preoperative diagnosis of chronic knee PJI was made when the patient had pain in the knee and/or fistula, erythrocyte sedimentation rate > 30 mm/h and serum concentration of C-reactive protein (CRP) > 1.0 mg/dL, radiological signs of loosening, ^99^Technetium methylene diphosphonate scintigraphy and ^99^Technetium hexamethylpropylene-amine oxine-labelled leukocytes scintigraphy were positive for infection, and when a culture of synovial fluid obtained by joint aspiration was positive. The definitive diagnosis of infection of the revision was determined using the criteria proposed by the Musculoskeletal Infection Society [23]. We did not use the criteria of Parvizi et al. [3], as they were not yet described in the literature at the time of the surgery. Items such as D-dimer, alpha-defensin, synovial CRP, and purulence were either not routinely measured at our center, or data were not collected (purulence). Intravenous broad-spectrum antibiotics (prolonged prophylaxis antibiotic) were started on the day of the surgery and maintained for 5–10 days. The oral switch was performed according to the antibiogram. Intravenous plus oral antibiotic therapy lasted a maximum of 6 weeks. In patients with a positive preoperative culture for coagulase-negative staphylococci (CoNS) or with a negative culture, intravenous broad-spectrum antibiotics were started on the day of the surgery. Patients with a preoperative culture positive for any microorganism other than CoNS were given an oral or intravenous antibiotic 25 days pre-operatively, according to the antibiogram test. In patients with suppressive treatment and a known microorganism, this treatment was not withdrawn before surgery. After discharge, the patients were followed up at 3 months, 6 months, and 1 year. At each visit, clinical response and adverse events were recorded. Outcomes were classified as follows after the final visit: (1) cure—when the patient presented a good functional arthroplasty with no local signs of infection and a value of CRP < 1.0 mg/dL, or (2) failure—when these criteria were not met. Classification as failure implied the need for taking out the prosthesis implanted during the 1-stage revision or for suppressive antibiotic treatment. Acute debridement after 1-stage revision arthroplasty was considered a complication, not a failure. In the last visit, functional results were determined according to the Knee Society Score (KSS).

### 2.2. Prolonged Prophylaxis Antibiotic, Intraoperative Microbiology, and Histology

Patients starting prolonged antibiotic prophylaxis on the surgery day received the antibiotic half an hour before the procedure. At the time of prosthesis removal, samples for the microbiological study were always taken; at least six periprosthetic samples from different sites were submitted to the laboratory for culture. Samples for the histological study were obtained from the periprosthetic membrane around the tibial and femoral parts. The Pathology Department at our hospital follows Mirra’s criteria (adapted by Feldman) for infection, considering a positive result when five neutrophils per high-power field (400×) are found in at least five separate microscopic fields.

### 2.3. Surgical Procedure

The surgery was split into two parts. The patient was placed in a supine position, and removal was always performed through pre-existing incisions. Meticulous surgical debridement was performed, and the wound was washed out with 9 L of normal saline. This was the end of the first part of the operation, and the patient was redraped and new instruments were used. The surgical team rescrubbed and put on new gowns. In all cases, implantation of a new arthroplasty was performed. Reconstruction of the joint was carried out with implantation of a cemented knee implant (eight cases of Endo-Model^®^ Rotating Hinge Prosthesis Link^®^ (Hamburg, Germany), one case of Megasystem-C Revision System Link^®^ (Hamburg, Germany), eight cases of Legion RK Revision Knee Replacement Smith&Nephew^®^ (Hertfordshire, United Kingdom), one case of Legion HK Hinge Knee Replacement Smith&Nephew^®^ (Hertfordshire, United Kingdom), and one case of P.F.C.^®^ Sigma^®^ TC3™ DepuySynthes^®^ (Warsaw, IN, USA). Antibiotic-loaded (gentamicin or gentamicin plus vancomycin) polymethylmethacrylate (PMMA) bone cement was used for both the fixation of the new implant and reconstruction of bone defects. The surgical team used premixed antibiotic bone cement and we never added any other antibiotics into the cement.

### 2.4. The Principal Variables

There were two main variables in this study: (1) infection control, defined by the number of patients cured/number of patients on who a 1-stage revision had been performed, and (2) economic costs. The costs were calculated using average cost in USD as described by Srivastava et al. [17] for 1-stage or 2-stage revisions. Costs calculated by Srivastava et al. [17] included the total fee of the surgical procedure, anesthesia costs, implant costs, hospital stay, and professional fees. These authors calculated costs based on USD.

## 3. Results

Thirty-nine patients were preoperatively diagnosed with chronic knee PJI from September 2015 to December 2017. The mean, median, and minimum follow-up was 43.8 months, 43 months, and 26 months, respectively. A 2-stage revision was performed on 16 patients (five patients who came from an acute infection in which the recent DAIR had failed, four patients with the presence of systemic sepsis or large abscesses in the calf or along the entire limb, four patients with significant skin defects who needed plastic surgery, two patients with severe damage or rupture of the extensor mechanism, and one patient with fungal infection), and they were excluded from the study (Table 1). A 1-stage revision was performed in five patients with preoperative diagnosis of chronic knee PJI, but infection was not confirmed in the intraoperative findings. In these patients, the antibiotic was stopped after 5–10 days of intravenous broad-spectrum antibiotics. These patients were also excluded from the study (Table 1). After exclusions, 18 patients were included in the study. Characteristics of the patients included in the study and preoperative findings are shown in Table 2, and intraoperative and postoperative findings are shown in Table 3.

All 18 cases fulfilled the infection criteria described by the Musculoskeletal Infection Society, 17 cases fulfilled criterion 1 (a pathogen is isolated by culture from at least 2 separate tissue or fluid samples obtained from the affected prosthetic joint) [23], and one case fulfilled criterion 2 (sinus tract) [23]. Sixteen patients presented with positive joint fluid culture in arthrocentesis prior to surgery, and one case presented with a fistula with positive cultures of fistula fluid. In these cases, the infection was already confirmed preoperatively, either by positive culture or by the presence of fistula (Table 2). Only one case (number 18) in which preoperative joint fluid culture was negative was the infection confirmed with intraoperative cultures (Table 3). Patients who had microorganisms other than coagulase-negative staphylococci (CoNS) received antibiotic treatment before surgery to decrease the bacterial load on the soft tissues and avoid spreading the infection at the time of 1-stage replacement. Patients with CoNS who received antibiotic treatment before 1-stage replacement did so because they were on suppressive treatment (cases 1, 3, and 16) (Table 2). Furthermore, there were six surgical complications in five patients: two acute debridements, one aseptic revision due to instability, one tendon repair, one closed reduction, and one neuroma excision.

Nine patients were cured, albeit with a poor postoperative functional scores (Table 4). Most of these nine patients had poor outcomes due to stair climbing, with walking performance that was acceptable when considering preoperative levels. There was an overall improvement of the KSS scores in cured patients. The functional score decreased from preoperative levels in two patients (2 and 4), although the decrease is not ascribed to the surgery. One of these cases showed a pathological worsening (patient number 2; multiple new vertebral fractures), while in the other, pre-existing diseases were aggravated (patient number 4; diabetes mellitus, hepatopathy, and chronic obstructive pulmonary disease). All patients that were treated improved their knee score except patient number 2, who suffered multiple new vertebral fractures. All patients began walking with two sticks within 24–48 h of surgery and making flexo–extension movements of the operated leg. Patients’ outcomes as shown by knee score are given in Table 3. Thus, from a functional point of view, the 18 patients were spared from having to carry a spacer, from an interim period between two surgeries with a spacer, and from a second surgery for placement of a definitive prosthesis.

The cost for treating these 18 patients using 1-stage revision, according to the cost criteria described by Srivastava et al. [17], totaled USD 435,186 (18 patients × USD 24,177 per patient). Added to this cost are the costs of the six surgical complications occurring in five patients: two acute debridements (USD 17,723 per patient = USD 35,446), one aseptic revision (USD 24,177 per patient), one tendon repair (USD 24,177 per patient), one closed reduction (USD 24,177 per patient) (Figure 1 and Figure 2), and one neuroma excision (USD 24,177 per patient). The total cost for treating these 18 patients using a 1-stage revision amounted to USD 567,340. If these patients had been treated using 2-stage revision, the costs would have totaled USD 858,492 (18 patients × USD 47,694 per patient) without any complications. Therefore, the total economic savings for our hospital for these 18 patients was USD 291,152.

## 4. Discussion

Preliminary results after introducing 1-stage revision arthroplasty for the treatment of chronic knee PJI in lieu of the 2-stage procedure were promising from both a clinical and economic point of view. Our cure rate is currently 94.4%. If this level is maintained for these patients, and for future patients to be included in this study, then a 1-stage revision for infected TKA could become the elective option. With a 1-stage revision, there were fewer medical complications and functional recovery was faster as a consequence of fewer surgeries being performed. In addition, significant savings were realized. Recently, Parvizi et al. [24] indicated that treatment failure following 2-stage revision arthroplasty is high. No significant difference in treatment failure following the 2-stage revision arthroplasty was found between the 2000–2010 cohort and the 2011–2016 cohort. Therefore, novel treatments and techniques are certainly needed. Our results show there are great gains to using 1-stage revision arthroplasty for the treatment of chronic knee PJI [18].

While 1-time revision is the elective surgery for chronic hip PJI [18], it was not always possible to perform this procedure on chronic knee PJI. This fact implies a large selection bias, as the more complex cases of chronic infection are not included in our study. We encountered several cases in which the procedure could not be used:(1)Patients who came from an acute infection with a failed recently performed DAIR (5 cases out of 39). Although we could have classified these patients as chronic PJI, in our hospital these patients were considered a special subgroup within the chronic PJI, both hip and knee, because of their peculiarities [18]. Following DAIR failure, the recommendation for subsequent treatment was often a 2-stage revision arthroplasty [25,26,27]. Individuals with a DAIR failure inherently have a higher risk of failing a subsequent 1 or 2-stage revision arthroplasty [25,26,27]. These are patients who have recently undergone at least two surgeries, prosthesis implantation surgery and DAIR, before developing chronic PJI. This implies that they are patients who cannot be optimized preoperatively, as opposed to patients with chronic infection occurring months or years after the first surgery [28]. In addition, patients with a previous DAIR failure had recently received various antibiotic treatments, many of the broad-spectrum types, for the treatment of acute PJI. This led to having to consider changes in skin flora, and the appearance of more resistant organisms, or else we would not be able to identify which was the microorganism responsible for the chronic infection [29]. More importantly, we were left without options for identifying the causative microorganism. In most cases of failed DAIR, the chronic infection was not caused by the same microorganism causing the acute infection [29], and we were left without options to identify it. This was because at the time of failure and at the time of single-stage replacement, the patient had been under prolonged antibiotic treatment [29]. Therefore, when undergoing a 1-stage revision in this type of patient, we were faced with either of two situations, i.e., unclear pre-operative bacterial specification or non-availability of appropriate antibiotics. These are precisely the contraindications described by Gerke et al. [30] or the International Consensus Meeting (ICM) 2013 [31] to performing a 1-stage revision.(2)The presence of generalized sepsis or a large abscess in the leg. This was also a major cause of exclusion in our series (4 cases out of 39). Two patients had generalized sepsis, one patient had an involvement of the entire area of the calf with a fistula on the back of the leg, and one patient had a collection along the lateral face of the femur and thigh down the entire osteosynthesis plate, which led to a periprosthetic fracture. In these patients, it is common sense and has also been described by Lichstein et al. [31] not to perform a 1-stage revision, and the patient’s life has to be saved through the 2-stage revision.(3)The presence of severe soft tissue deficiency over the joint. This was also a major cause of exclusion in our series (4 cases out of 39), and a clear difference between the hip series [18] and the current knee series. In the hip series, no patient was excluded because of a skin defect, while in the knee series it was a fairly common situation. In these patients, a 2-stage revision was a better option than a single-stage revision, for two reasons: firstly, because any soft tissue defects can be fixed initially, along with a cement spacer [32], and, secondly, because patients who require plastic surgery for a skin defect often need more than one surgery [33]; a 1-stage revision would imply re-exposing the final prosthesis to the risk of a new infection. The presence of severe soft tissue deficiency over the joint is a contraindication described by the University College London Hospital (UCLH) [8], the Infectious Diseases Society of America (ISDA) [22], and a relative contraindication described in the ICM 2018 [34].(4)Severe damage or rupture of the extensor mechanism. This was a less important cause of exclusion in our series (2 cases of 39). For a patient with severe damage or rupture of the extensor mechanism plus the presence of infection, the indicated treatment would be an arthrodesis, with a fixed spacer or directly fixed with a nail [30].(5)Fungal infection. This was a minor cause of exclusion in our series (1 case of 39). Fungi have always been a very rare cause of chronic PJI [34], and the presence of a fungus has always been considered a contraindication for 1-stage revision [2,21,22], although there are current series with good results using 1-stage revision [35]. In our case, as we were beginners to the 1-stage revision in TKA, we were conservative and chose the 2-stage revision procedure. We believe that we will soon have to perform the 1-stage revision in cases with a fungal infection, as also seen in the literature [35]. We did not exclude any patient presenting with a resistant microorganism or pre-operatively with an unknown microorganism. There are authors who contraindicate 1-stage revision in case of infection by Gram-negative bacilli [2,21,22]. We included two patients with Gram-negative bacilli (*Klebsiella pneumoniae* and *Escherichia coli*). Recently, Citak et al. [36] described that *Enterococcus* and *Streptococcus* species were associated with a higher risk of failure after 1-stage exchange arthroplasty. In our series, there were five patients (27%) with *Streptococcus* species, and the evolution of all of them has been good. The difference between our patients with *Streptococcus* species and those in the Citak et al. [36] series is that our patients received antibiotic treatment prior to surgery (Table 2).

Our study followed a systemic and local antibiotic protocol that was quite different from other 1-stage revision protocols [8,30,37,38]; (1) in several studies, single-stage replacement and intraoperative infusion of local antibiotic with a catheter for a few days [37] were performed concurrently. In our study, this system was not used. We believe that chronic knee PJI is initiated when a biofilm forms over the implant, and, therefore, removing an implant while performing an ample debridement and applying appropriate antibiotic coverage at this stage allows the patient to be cured. (2) An intravenous broad-spectrum antibiotic (prolonged antibiotic prophylaxis) was started on the day of surgery and maintained for 5–10 days if the microorganism causing the infection was an ECN or was unknown, or an oral or intravenous antibiotic was initiated 2–5 days earlier according to the antibiogram test (if different from ECN). Infections due to an ECN are considered easily treatable because they are low-virulence microorganisms and the bacterial load in periprosthetic tissues is low; the large majority of bacteria adhere to the implant. Thus, the removal of the implant and an intravenous broad-spectrum antibiotic initiated on the day of surgery (prolonged antibiotic prophylaxis) plus oral antibiotic is sufficient to cure the infection. When the microorganism is not identified preoperatively, this strategy is also applied for identifying the causative microorganisms (e.g., patient number 18). In infections that are due to a microorganism other than ECN, an oral or intravenous antibiotic is initiated 2–5 days preoperatively according to the antibiogram test in order to decrease the bacterial load in periprosthetic soft tissues and to ensure the success of implant removal, a wide debridement, and appropriate antibiotic coverage. Most authors begin antibiotic treatment on the same day of surgery [8,30,38] and only use the 2–5 preoperative-days of antibiotic treatment when specific microorganisms have been previously identified as the cause of the chronic PJI. (3) Authors select the antibiotic to be added in cement depending on the susceptibility of the microorganism [30], or even contraindicate 1-stage revision if the microorganism is unknown because they attribute the local antibiotic part of the treatment of this chronic infection [34]. We have always used a fixed antibiotic in cement, similar to other authors [38]. The concept is that the local antibiotic in cement is a prophylactic for new infections; it does not serve as treatment for the chronic infection [39], or for an acute infection during the 1-stage revision [40].

The clinical results for 2-stage reviews have always shown a success rate of over 90%. However, the current series has shown us that this success rate is not accurate because it does not consider the patients with reimplantation failure or mortality [24]. According to a recently published experience of 1-stage and 2-stage revisions [24,36], we need to re-evaluate which is the best surgical approach for managing chronic hip prosthesis infections. An appropriate treatment for chronic knee-arthroplasty infection could be the 1-stage revision itself [17]. Gulhane et al. [41] and Gehrke et al. [30] already suspected this in their latest articles, entitled “Single stage revision: regaining momentum” and “One-stage exchange: it all began here”, respectively. Our study is another good example of the successful application of 1-stage revision. Recently, there have been several systematic reviews and meta-analyses that have compared 1-stage revision to 2-stage revision and all have come to quite similar conclusions: 1-stage revisions have a very similar success rate as 2-stage revisions, along with all that is implied. The systematic revision or meta-analysis carried out by Kunutsor et al. [12] concluded that the rate of re-infection was 7.6% (3.4–13.1) in 1-stage studies and 8.8% (7.2–10.6) in 2-stage studies. Nagra et al. [15] carried out a systematic revision of studies with 1- or 2-stage revision arthroplasty, with more than ten patients with a minimum 2-year follow-up. They concluded that recent studies suggest 1-stage revision arthroplasty may provide superior outcomes, including lower reinfection rates and superior function, in select patients. Chew et al. [10] conducted a systematic review of 1-stage revision studies between 1985 and 2015 and found that infection rates were lower in post-2000 studies; reinfection rates in current studies on 1-stage revision in TKA are also lower. Srivastava et al. [17] performed decision analysis to determine the optimal PJI management following TKA using the quality-adjusted life-years (QALYs) dimension and economic costs. The conclusion was that the optimal surgical course producing the highest quality of life was a 1-stage revision. However, the 2-stage revision is currently considered the gold standard for infection eradication in patients with PJI following TKA. This is because the QALYs were based on death rates, failure to have a second stage reimplantation, the utility value associated with the time between stages, and the time-dependent utility effect of a longer recovery [42]. Therefore, it appears that the 1-stage revision TKA has the potential to be more beneficial to patients than the 2-stage revision [10].

An inevitable limitation of our study was that follow-up time was shorter than desirable (2–4 years). Moreover, few patients could be included in the study. Nevertheless, Xu et al. [43] recently suggested that a one-year follow-up is sufficient for an accurate reporting of treatment failure. In any case, our results are relevant from the point of view of the morbidity rates associated with surgery and the patient quality of life, i.e., we were able to cure infection with a single surgical intervention, and patients had a normally functioning knee prosthesis allowing them to carry out their day-to-day activities. Two of the 18 patients were operated on due to infection. In one patient, it was necessary to perform a DAIR because of acute infection. This patient had a history of two 2-stage revisions due to PJI. Recently, Citak et al. [36] pointed out that a history of 2-stage exchange due to PJI is a risk factor associated with reinfection following 1-stage exchange TKA. In one patient, it was necessary to perform a DAIR and suppressive treatment to control the PJI. Four of the 18 patients were operated on due to mechanical complications (i.e., instability, dislocation, tendon rupture, and neuroma). Another drawback of our study was that economic costs were calculated according to Srivastava et al. [17]. It would have been preferable to calculate the real costs of 1- and 2-stage revision arthroplasty in our institution. However, these data were not available. Nevertheless, we deemed that using the costs as given by another author provides a sufficiently good approximation of the actual economic savings. In addition, Srivastava et al. [17] performed a cost analysis by calculating direct costs as a global fee of the surgical procedure, anesthesia costs, implant costs, hospital stay, or professional fees, but did not include indirect costs, such as social costs (help from family members, time spent in residencies in the meantime, etc.). Therefore, the overall economic savings of 1-stage revision arthroplasty as against the 2-stage procedure could be even greater than estimated by Srivastava et al.

## 5. Conclusions

In conclusion, applying a 1-stage revision for chronic TKA with minimal exclusion criteria can result in a high clinical success rate and a significant reduction in morbidity/mortality compared to the 2-stage surgical option. This clinical success has led to a major change in treatment strategies for TKA at Hospital Clínic de Barcelona, in addition to substantial economic savings.

It is important to continue follow-up of these patients and to include new patients and results with an aim to further support these preliminary results.

## Figures and Tables

**Figure 1 antibiotics-10-01436-f001:**
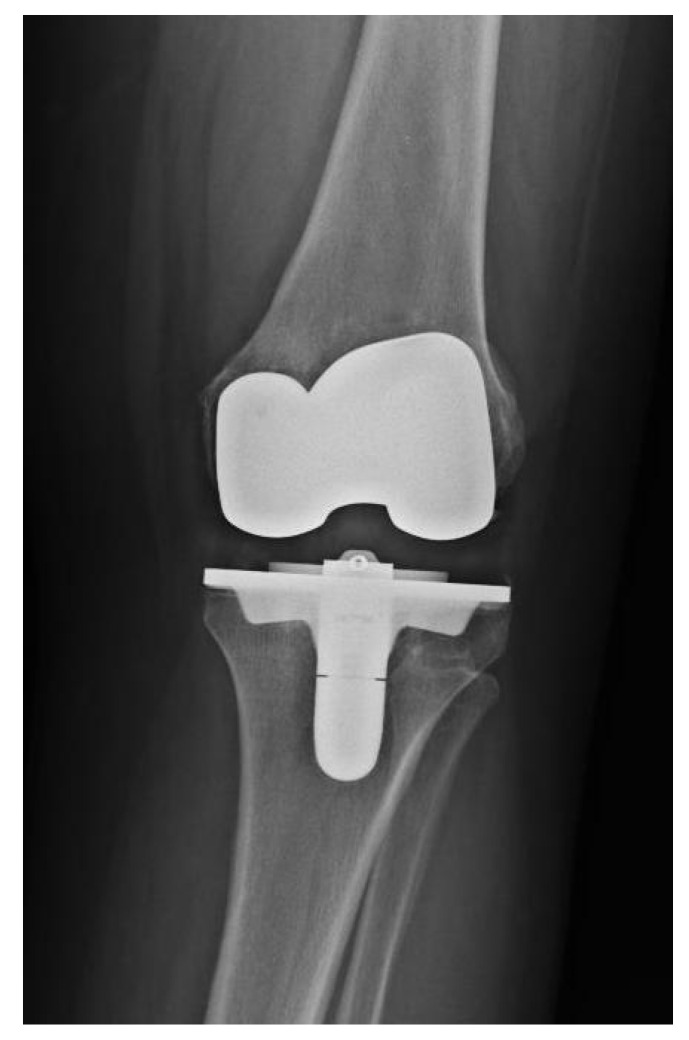
Preoperative X-ray of knee joint with *Streptococcus anginosus* infection of patient number 5.

**Figure 2 antibiotics-10-01436-f002:**
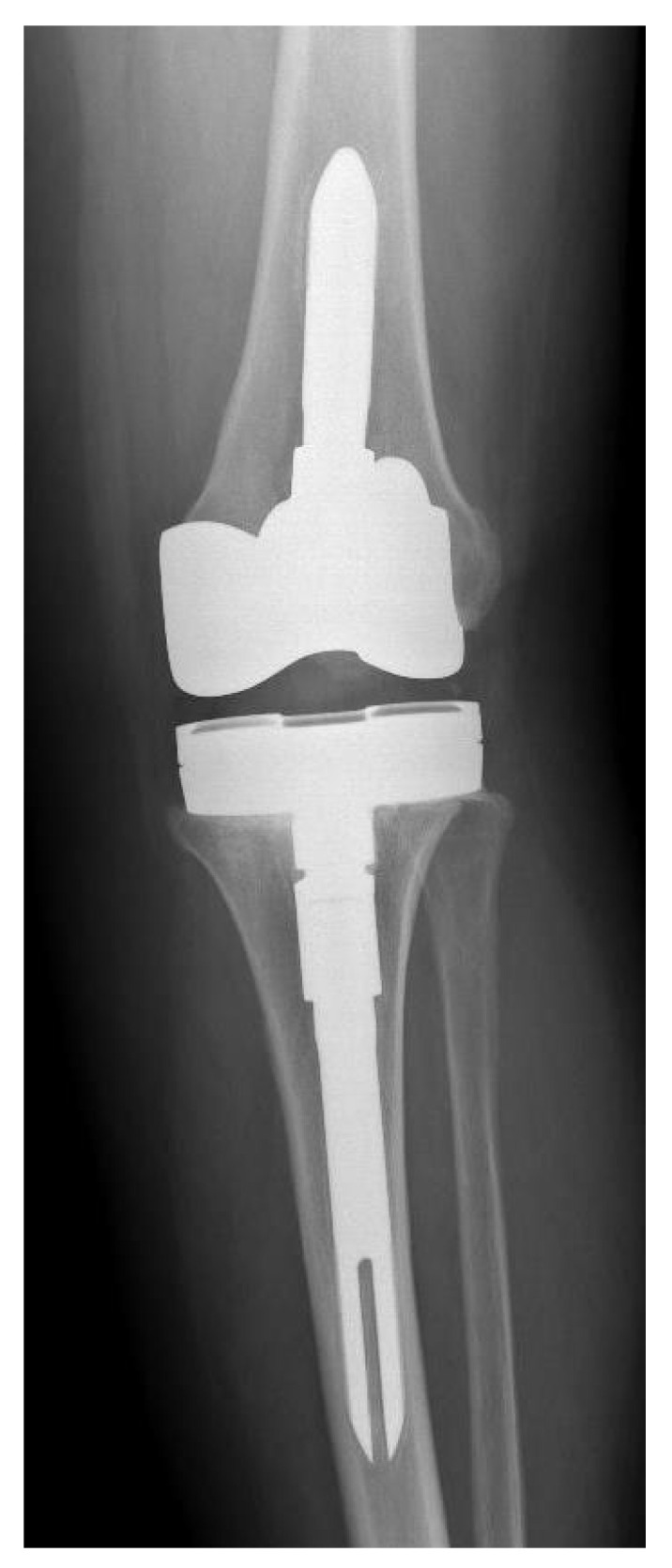
Postoperative X-ray of new knee prosthesis of patient number 5.

**Table 1 antibiotics-10-01436-t001:** Patients included and excluded in the study.

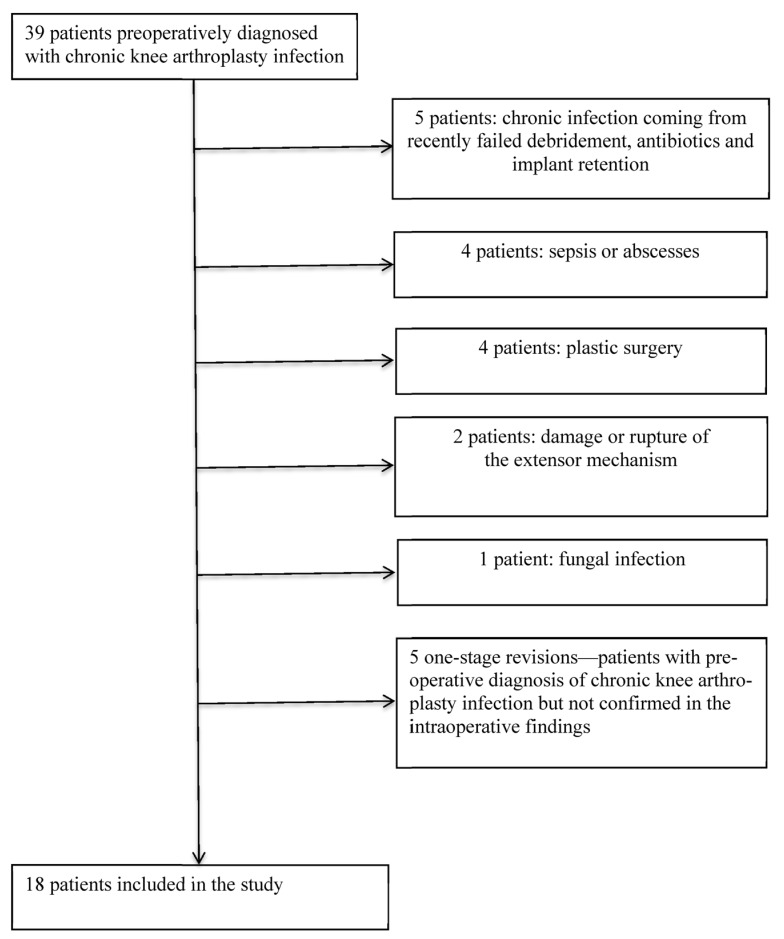

**Table 2 antibiotics-10-01436-t002:** Characteristics of patients and preoperative findings.

N	Age/Gender/ASA	Comorbidities	Primary Diagnosis	Age of the Prosthesis (Years)	CRP/ESR	Signs of Radiological Loosening/Sinus Tract	Bone Scintigraphy	Leukocyte Scintigraphy	Arthrocentesis Synovial WBC/Synovial PMN (%)	Arthrocentesis Organism	Previous Antibiotic	Previous Type of Antibiotic
1	84/W/II	0	Osteoarthritis	0.9	2.19/74	Yes/Yes	NP	NP	NP/NP	*Staphylococcus* *epidermidis*	Yes (suppressive)	Cotrimozazole ^c^
2	75/W/III	3	Osteoarthritis	4	0.33/3	Yes/No	NP	NP	NP/NP	*Staphylococcus* *epidermidis*	No	-
3	67/M/II	1	Osteoarthritis	14	2.2/2	Yes/Yes	NP	NP	NP/NP	*Staphylococcus* *epidermidis* ^b^	Yes (suppressive)	Levofloxacin + Minociclyn ^d^
4	62/W/III	4,5,6	Osteoarthritis	1.7	4.9/24	Yes/No	NP	NP	34.860/95	*Staphylococcus* *lugdunensis*	No	-
5	72/W/II	1	Osteoarthritis	5	8.06/107	Yes/No	Loosening	Infection	37.400/90	*Streptococcus* *anginosus*	Yes	Amoxicillin
6	71/W/II	1	Osteoarthritis	9	11.79/131	Yes/No	Loosening	NP	NP/NP	*Streptococcus* *salivarius*	Yes	Amoxicillin
7	81/W/II	1	Osteoarthritis	4	3.3/92	Yes/No	NP	NP	NP/NP	*Staphylococcus* *epidermidis*	No	-
8	81/M/II	1	Osteoarthritis	0.9	8.48/100	Yes/No	NP	NP	57.880/96	*Klebsiella* *pneumoniae*	Yes	Ceftriaxone (6 days)
9	69/M/II	1,4	Osteoarthritis	1.5	2.23/5	Yes/Yes	Loosening	Infection	NP/NP	*Staphylococcus* *epidermidis*	No	-
10	84/W/III	1,2,3	Osteoarthritis	3	14.1/41	Yes/No	NP	NP	NP/98	*Clostridium* *histolyticum*	Yes (suppressive)	Amoxicillin/clavulanic
11	68/W/I	0	Osteoarthritis	2	3.91/126	Yes/No	NP	NP	23.360/87	*Streptococcus grupo viridans*	Yes	Amoxicillin
12	82/W/II	0	Osteoarthritis	5	1.54/43	Yes/No	Loosening	Infection	NP/NP	*Staphylococus hominis*	No	-
13	78/W/III	1,3,4	Osteoarthritis	0.8	15/140	Yes ^a^/No	NP	NP	50/NP	*Staphylococcus* *epidermidis*	No	-
14	79/W/II	1,3,4	Osteoarthritis	9	0.26/13	Yes/No	NP	NP	190/NP	*Streptococcus parasanguis/Streptococcus oralis*	Yes	Amoxicillin
15	87/W/III	1,2,3	Osteoarthritis	11	4.96/82	Yes/No	NP	NP	900/90	*Escherichia coli*	Yes (suppressive)	Ciprofloxacin
16	82/W/III	1,2,3	Osteoarthritis	3	7.52/59	Yes/No	Loosening	NP	26.750/90	*Staphylococcus* *lugdunensis*	Yes (suppressive)	Levofloxacin + Minociclyn
17	93/M/III	1,7	Osteoarthritis	3	16.2/-	Yes/No	NP	NP	950/NP	*Streptococcus gordonii*	Yes	Daptomycin + Ceftriaxone (6 days)
18	69/M/II	1,2,3	Osteoarthritis	4	0.28/5	No/No	Loosening	Infection	740/28	Negative	No	*-*

N: number of patients; Comorbidities: 1: High blood pressure, 2: Chronic anticoagulation, 3: Heart diseases, 4: Diabetes mellitus, 5: Hepatopathy, 6: Chronic obstructive pulmonary disease, 7: Chronic kidney disease; Gender: W: women, M: men; CRP: C-reactive protein (mg/dL), ESR: erythrocyte sedimentation rate (mm/h); WBC: white blood cell; NP: not performed; ^a^ Infected femur pseudoarthrosis + loosening knee arthroplasty; ^b^ From sinus tract; ^c^ Cotrimoxazole 2 months before + (Linezolid + Cotrimoxazole) 3 weeks before + (Meropenem + Daptomycin) 2 days before; ^d^ (Levofloxacin + Minociclyn) one week a month.

**Table 3 antibiotics-10-01436-t003:** Intraoperative findings, antibiotic therapy, complications, and outcomes.

N	Histology	L	Sm	Sc	Microorganism	ArthrocentesisSynovial WBC/Synovial PMN (%)	Intravenous Prolonged Prophylaxis Antibiotic (5–10 days)	Cement Antibiotics/Oral Antibiotics/Days with Intravenous and Oral Antibiotics	Complications	Follow-Up (Months/Last CRP/Cured or not Cured)	Knee Society Score (Preoperative/Last Follow-Up)
1	Positive	0/2	0/2	0/2	Negative	850/83	Meropenem + Daptomycin	G/Tmx + R/42	No	61/<0.4/Cured	57–30/94–40
2	Negative	0/2	0/2	0/2	Negative	50/NP	Meropenem + Linezolid	G + V/Lin/42	Instability ^d^	55/<0.4/Cured	66–50/50–0 ^k^
3	Positive	0/2	0/2	0/2	Negative ^a^	NP/NP	Meropenem + Linezolid	G/Lin + Amox/42	No	52/<0.4/Cured	63–50/94–80
4	Negative	2/2	0/2	0/2	*Staphylococcus* *lugdunensis*	NP/94	Meropenem + Cloxacillin	G/Le + R/42	No	50/<0.4/Cured	57–25/69–10
5	Negative	0/2	0/2	0/2	Negative	37,580/94	Meropenem + Linezolid	G/Amox + R/42	Dislocation at 9 weeks ^e^	49/<0.4/Cured	53–15/94–45
6	Negative	0/2	0/2	0/2	Negative ^b^	24,830/91	Meropenem + Linezolid	G/Amox + R/42	No	48/<0.4/Cured	53–50/94–80
7	Positive	2/2	0/2	0/2	*Staphylococcus epidermidis*	31,200/97	Meropenem + Linezolid	G + V/Lin/42	No	47/0.8/Cured	55–50/64–55
8	Negative	0/2	0/2	0/2	Negative	21,120/86	Meropenem + Linezolid + Cipro	G/Ci/42	No	45/<0.4/Cured (Death)	−/94–55
9	Positive	2/2	0/2	1/2	*Staphylococcus* *epidermidis*	NP/NP	Meropenem + Linezolid	G/Lin + Ci/42	Acute infection at 1 month/stiffness 0–45° ^f^	44/0.7/Cured	28–5/51–25
10	Positive	0/2	0/2	0/2	Negative	NP/NP	Meropenem + Linezolid	G/Moxi/42	No	42/2.5 ^j^/Cured	53–5/96–80
11	Positive	0/2	0/2	0/2	Negative	25,000/89	Meropenem + Linezolid	G/Amox + R/42	No	42/0.8/Cured	43–60/95–100
12	Negative	0/2	0/2	0/2	Negative	NP/NP	Meropenem + Linezolid	G/Le + R/42	No	41/1.6/Cured	51–0/88–60
13	Negative	0/2	0/2	0/2	Negative	190/NP	Meropenem + Linezolid	G/Le + R/42	Quadricipital tendon rupture at 2 months/relapse at 26 months ^g^	26/–/Failure	44–0/–
14	Negative	0/2	0/2	0/2	Negative	1280/NP	Meropenem + Linezolid	G/Le + R/42	No	39/<0.4/Cured	53–30/89–60
15	Positive	0/2	0/2	0/2	Negative	NP/95	Meropenem + Linezolid	G/Ci/42	No	38/<0.4/Cured	43–0/94–60
16	Positive	0/2	0/2	0/2	Negative	NP/NP	Meropenem + Linezolid	G/Le + R/42	No	36/<0.4/Cured	55–0/94–15
17	Negative	0/2	0/2	0/2	Negative	12,800/79	Meropenem + Linezolid	G/Amox + R/42	Periprosthetic fracture at 29 months ^h^	37/<0.4/Cured	−/88–50
18	Negative	0/2	1/2	1/2	*Staphylococcus lugdunensis/Staphylococcus epidermidis ^c^*	NP/51	Meropenem + Linezolid	G/Dal/42	Neuroma of infrapatellar branch of saphenous nerve ^i^	37/0.48/Cured	65–30/74–70

N: Number of patients; Histology: Positive: ≥5 neutrophils per high-power field (400×), Negative: <5 neutrophils per high-power field (400×), NP: Not performed; L: liquid samples inoculated in blood culture flasks, S: solid samples; Sw: swab samples; ^a^
*Propionibacterium acnes* in sonication; ^b^
*Streptococcus vestibularis* in sonication; ^c^
*Staphylococcus epidermidis* in sonication also; ^d^ The patient needed to change the polyethylene 8 months later; ^e^ The patient needed a closed reduction ^f^ The patient needed an irrigation and debridement ^g^ The patient needed a tendon repair. Irrigation and debridement plus suppressive antibiotic treatment;^h^ Orthopedic treatment with plaster was used; ^i^ Pain due to the infrapatellar branch of saphenous nerve. Excision of the neuroma was performed 14 months later; ^j^ Polymyalgia rheumatic; ^k^ Multiple vertebral fractures; CRP: C-reactive protein (mg/dL); Antibiotic-loaded cement: G: gentamicin, V: vancomycin; Oral Antibiotics: Le: levofloxacin, R: rifampicin, Lin: linezolid. Ci: ciprofloxacin, Amox: amoxicillin; Tmx: trimetropim–sulfamethoxazole, Moxi: moxifloxacin, Dal: dalbavancin.

**Table 4 antibiotics-10-01436-t004:** Infection control and Knee Society Score (KSS) results.

Infection Control	Scores	Preoperative Results		Postoperative Results	
		Knee Score	Functional Score	Knee Score	Functional Score
Cure	Excellent	-	-	11	4
	Good	-	-	1	1
	Fair	3	1	3	3
	Poor	12	14	2	9
Failure	Excellent	-	-	-	-
	Good	-	-	-	-
	Fair	-	-	-	-
	Poor	1	1	-	-

Scores; Excellent: 100–80, Good: 70–79, Fair: 60–69, Poor: <60.

## Data Availability

The datasets used in the current study are available from the corresponding author on reasonable request.

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
