# Peer review of "Experiences during Switching from Two-Stage to One-Stage Revision Arthroplasty for Chronic Total Knee Arthroplasty Infection"

_antibiotics, 2021, doi:10.3390/antibiotics10121436_

Round 1
Reviewer 1 Report
I have previously rejected the manuscript due to insufficient English language and a poor study design. I have once again read through the manuscript and found several language mistakes (e.g. l 78 "fungi infection" vs. "fungal infection"; l 131 "supinion position" vs. "supine position"). Thus, I am not able to change my recommendation for the editors but I am happy to step back from my reviewership so that the editor can invite another reviewer. Good luck!
Author Response
REVIEWER 1
I have previously rejected the manuscript due to insufficient English language and a poor study design.
I can come to understand the reviewer, as at first glance it seems like a series of 18 cases, but I’m a little disappointed with the same that he doesn’t appreciate the things, from my point of view, that are interesting from the study:
1.- That in this article we explain how an entire trauma or hospital service changes its surgical strategy after 15 years using a 2-stage revision arthroplasty. It is not at all easy to take this step, as the treatment of prosthetic infection has a multidisciplinary approach, and making such a change implies that you have to change the habits of many people, one of the great difficulties we encounter in the medicine.
2.- It is a study that could be done because the patients were collected prospectively, and we know the reason why the replacement was or was not done in 1-stage revision arthroplasty.
3.- There is an antibiotic treatment strategy in order to perform the 1-stage revision arthroplasty that is poorly described or not at all described in the literature, which from my point of view is very interesting.
Apart from these facts, or whether or not the article is accepted for publication, in the future if the publisher of the journal deems it appropriate, and the reviewer wants, I would like to be able to discuss the article with the reviewer, as I am interested and passionate about 1-stage revision arthroplasty and all that it means. Just for the time that the reviewer has devoted to my article I have to thank him.
I have once again read through the manuscript and found several language mistakes (e.g. l 78 "fungi infection" vs. "fungal infection"; l 131 "supinion position" vs. "supine position").
We agree with the reviewer and we correct again all manuscript by an expert translator.
Thus, I am not able to change my recommendation for the editors but I am happy to step back from my reviewership so that the editor can invite another reviewer. Good luck!
Reviewer 2 Report
The authors have address all concerns
Author Response
REVIEWER 2
The authors have address all concerns.
Thank you.
Reviewer 3 Report
Lines 60-66: from my point of view, the literature reproduction does not belong in this section of the article Line 114: how were cases dealt with (which category were they assigned to) that showed poor functional results without infection? Line 131: supinion means supine? Line 141: with which antibiotic was the bone cement fortified? Line 144: the 2-stage surgical concept should also be mentioned Line 149: the basic principles of the cost analysis should be set out Line 151: it is confusing when the author first states that up to Sept 2015 all procedures were carried out in two stages and subsequently in one stage, but then outlines 2stage interventions from September 2015 Table 2: could the author list the types of comorbidities? Table 3: A short description of the clinical course apart from the table would certainly be advantageous, I also miss the direct comparison between the procedures A faster reconvalescence is described in the further course, which is not further specified Lines 220-283 should be shown much shorter, because this is not the essence of the study Line 301: has the preoperative antibiotic approach been evaluated, or is it more the author's evidence? In the entire discussion, only one sentence is lost about the economic advantages of the 1-stage procedure although it is a key point of analysisAuthor Response
REVIEWER 3
Lines 60-66: from my point of view, the literature reproduction does not belong in this section of the article.
I agree with the reviewer, that normally a review of the literature is not put into material and methods, but in this article this text is not a review of the literature, but are the reasons we used to change surgical strategy, and therefore we believe it is part of the material and methods (it is an important part of the work or study), because we want to explain to readers the cause or why of our change of strategy. In our case 4 situations came together. We want to explain the reality we lived in at that time. I think this part is very important, because it’s very difficult to make a change in surgical strategy when you’ve been operating in a way for 15 years, and you want to make a change.
We prefer to keep this part in this section.
Line 114: how were cases dealt with (which category were they assigned to) that showed poor functional results without infection?
I think this reviewer comment is very interesting. The article describes on the one hand whether the infection is cured or not, and in Table 3 the reader can see the functional results of the replacement prosthesis. In most cases we can see that the results of the KSS have improved. To highlight these functional results we have introduced in the results section a table 4 where we relate the fact that it is cured or not, and the functional state according to the following reference:
Asif S, Choon DSK. Midterm results of cemented Press Fit Condylar Sigma total knee arthroplasty System. J Orthop Surg (Hong Kong). 2005;13(3):280-4.
We also introduced a paragraph in result section.
Nine patients were cured, albeit with a poor postoperative functional score (Table 4). Most of these 9 patients had a poor outcome due to stair climbing, with walking performance that was acceptable when considering preoperative levels. There was an over-all improvement of the KSS scores in cured patients. The functional score decreased from preoperative levels in two patients (2 and 4), although the decrease is not ascribed to the surgery. One of these cases showed a pathological worsening (patient number 2; multiple new vertebral fractures), while in the other pre-existing diseases were aggravated (patient number 4; diabetes mellitus, hepatopathy and chronic obstructive pulmonary disease). All patients treated improved their knee score except patient number 2, who suffered multiple new vertebral fractures.
Line 131: supinion means supine?
We agree with the reviewer. It’s a mistake. It means supine. We changed.
Line 141: with which antibiotic was the bone cement fortified?
In 16 cases the antibiotic was Gentamicin and in 2 cases Vancomycin plus Gentamicin. We have included this information in material and methods and in the results in the table 3.
Line 144: the 2-stage surgical concept should also be mentioned
At this point we disagree with the reviewer. In the study, we excluded patients who underwent 2-stage revision arthroplasty; we do not believe that it is necessary to write in material and methods the 2-stage replacement protocol of our hospital.
If the article is not accepted for this reason, we have no problem including it if you ask us.
Line 149: the basic principles of the cost analysis should be set out
We agree with the reviewer. We included the basic principles of the cost analysis in the text (material and methods):
“Costs calculated by Srivastava et al [17] include the total fee of the surgical procedure, anaesthesia costs, implant costs, hospital stay and professional fees.”
Line 151: it is confusing when the author first states that up to Sept 2015 all procedures were carried out in two stages and subsequently in one stage, but then outlines 2stage interventions from September 2015
We agree with the reviewer. The text has been changed to clarify when and under what criteria was a 1- or 2-stage arthroplasty carried out. Until Septembre 2015 all surgeries had been done in two stages. From this date it was decided to try to perform a 1-stage procedure whenever feasible. However, this was not possible in all cases, as shown schematically in Table 1.
In material and methods we introduced a new sentence:
“After September 2015, all TKA in Hospital Clínic that were diagnosed with chronic PJI and not excluded for various reasons (Table 1) were treated by 1-stage revision.”
Table 2: could the author list the types of comorbidities?
We agree with the reviewer. There was a mistake in the list of the types of comorbidities. EPOC is chronic obstructive pulmonary disease in Spanish, and IRC is chronic kidney disease in Spanish. We changed it in the text of table 2.
“Comorbidities: 1: High blood pressure, 2: Chronic anticoagulation, 3: Heart diseases, 4: Diabetes mellitus, 5: Hepathopaty, 6: Chronic obstructive pulmonary disease, 7: Chronic kidney disease”
Table 3: A short description of the clinical course apart from the table would certainly be advantageous, I also miss the direct comparison between the procedures A faster reconvalescence is described in the further course, which is not further specified
I agree with the reviewer and in the results section we introduced a small text.
All patients began walking with 2 sticks within 24-48 hours of surgery and making flexo-extension movements of the operated leg. Patients’ outcomes as shown by knee score are given in Table 3. Thus, from a functional point of view, the 18 patients were spared from having to carry a spacer, from an interim period between two surgeries with a spacer and from a second surgery for placement of a definitive prosthesis.
Lines 220-283 should be shown much shorter, because this is not the essence of the study
I agree with the reviewer that this section might be a little shorter, but I disagree that it is not the essence of the study. I think it is a very important part of the study and for this reason I have not shortened it, as we have to explain it very well and we have to give the reasons to the readers in the discussion of why we excluded from the 1-stage revision arthroplasty some patients, when at first we had planned to operate on them all using the 1-stage revision arthroplasty strategy.
Line 301: has the preoperative antibiotic approach been evaluated, or is it more the author's evidence?
The reviewer is right that the sentence is a bit confusing. Perhaps it would be better to say:
“In infections that are due to a microorganism other than ECN, 2-5 days preoperatively an oral or intravenous antibiotic is initiated according to the antibiogram test in order to decrease the bacterial load in periprosthetic soft tissues, and ensure success removing the implant, a wide debridement and appropriate antibiotic coverage”.
The goal of this antibiotic treatment (Patients with a preoperative culture positive for any microorganism other than CoNS were given an oral or intravenous antibiotic 2-5 days pre-operatively, according to the antibiogram test), is to decrease the bacterial load in periprosthetic soft tissues, at no time is the intention of this treatment to cure the bone infection or remove the biofilm from the prosthesis preoperatively. We believe that this is one of the novelties of this study.
At first, it is an author's evidence, but analyzing tables 2 and 3 we can see that most patients who have received antibiotic treatment, periprosthetic soft tissue cultures have been negativized. Therefore here we have evidence that giving a short period of antibiotic before surgery decreases the bacterial load of periprosthetic soft tissues, and helps the 1-stage revision arthroplasty to be successful.
In the entire discussion, only one sentence is lost about the economic advantages of the 1-stage procedure although it is a key point of analysis
The reviewer is right that in the discussion there is only one sentence about costs, but this was so because the difference between 1-stage and 2-stage is so obvious that we didn’t discuss it.
On the recommendation of the reviewer we have introduced a paragraph in the discussion.
Another drawback of our study was that economic costs were calculated according to Srivastava et al. [17]. It would have been preferable to calculate the real costs of 1- and 2-stage revision arthroplasty in our institution. However, these data are not available. Nevertheless, we deemed that using the costs as given by another author provides a sufficiently good approximation of the actual economic savings. In addition, Srivastava et al. [17] does a cost analysis by calculating direct costs as global fee of the surgical procedure, anaesthesia costs, implant costs, hospital stay or professional fees, but does not include indirect costs such as social costs (help from family members, time spent in residencies in the meantime...). Therefore, the overall economic savings of 1-stage revision arthroplasty as against the 2-stage procedure could be even greater than estimated by Srivastava et al [17].
Round 2
Reviewer 3 Report
The authors have submitted a revision of the work, which from the current point of view is sufficient. The complaints were taken into account, so publication is recommended.Author Response
Reviewer 3
The authors have submitted a revision of the work, which from the current point of view is sufficient. The complaints were taken into account, so publication is recommended.
Thank you very much for your comments during the review period, which has improved the manuscript.
This manuscript is a resubmission of an earlier submission. The following is a list of the peer review reports and author responses from that submission.
Round 1
Reviewer 1 Report
This is an interesting analysis of cases, but the lack of scientific rigor prohibits conclusions. The major concerns are: 1) Lack of microbiology confirmation of deep infections (only 4 out of 18 patients had positive cultures). For PJI cases without a deep infection from a highly virulent pathogen (e.g. MRSA), the cost-effectiveness of 1-stage revision is well-known. 2) There appears to be gross selection bias, as the most challenging cases (16 out of 39) we excluded from the study. Thus, this manuscript does not offer more that case reports, which may be of value reporting in a specialty Journal.
Reviewer 2 Report
The authors report their preliminary results of changing their surgical strategy from a 1-stage to a 2-stage-revision to treat periprosthetic knee joint infections.
Unfortunately, the study has several flaws:
- English language is inusfficient:
e.g.:
- Line 16: "The objective of the present study (...)" instead of "The objective of the study here (...)" (colloquial english).
- Line 18: "(...) chronic knee periprosthetic joint infection." instead of "(...) periprosthetic knee joint infection."
- Line 18-21: makes no sense.
- ...
- Study design/structure of the manuscript:
- The number of patients included is low: of 39 patients with chronic periprosthetic knee joint infection, the authors have included only 18 patients.
- Mean/median and minimum follow-up are not stated.
- No a priori power calculation was performed.
Unfortunately, I have to recommend to reject the present study.